# Breathwork and holistic wellbeing: A protocol for a scoping review

**Fern Eleanor Beauchamp**[1], **Emily Bispo**[1], **Iolanda Costa Galinha**[1], **Andrew H. Kemp**[2]*

**1** Faculdade de Ciências Humanas, Universidade Católica Portuguesa, Lisbon, Portugal, **2** School of Psychology, Faculty of Medicine, Health, and Life Science, Swansea University, Swansea, United Kingdom

* a.h.kemp@swansea.ac.uk

## Abstract

Breathwork has great potential for supporting connections to self, others, and nature, laying the foundations for individual, collective, and planetary wellbeing. While reviews have summarised the impacts of a wide range of breathwork techniques on outcomes relating to individual wellbeing, past work has neglected broader contexts. Here we present a protocol for a scoping review to identify and summarise the available literature relating to a broader range of wellbeing domains, focused on connection to self (individual wellbeing), others (collective wellbeing), and nature (planetary wellbeing). In synthesising the literature, we will determine how and when holistic wellbeing might arise through breathing interventions. Guided by standardised guidelines for conducting scoping reviews, six electronic databases (PubMed, Scopus, Web of Science, APAPsycArticles, Psychology and Behavioural Sciences Collection and Cochrane), grey literature and reference lists of included literature will be searched. Using the Rayyan platform, two reviewers will independently screen title/abstracts after which the full texts of relevant articles will be reviewed based on predetermined criteria. Study details, breathing intervention type and wellbeing outcomes relevant to the research question(s) will be extracted. Numerical analysis of data items and descriptive qualitative analysis of data across the three domains will be reported. Preliminary results will be shared with breathwork facilitators, and their feedback will help highlight gaps not explored in the literature and to refine discussion of the findings in an applied context. We hope that results will inform subsequent research and encourage deeper reflection on the role of breathwork across wider contexts. This research will contribute to the growing evidence base for promoting the inner development of individuals and communities focused on major societal challenges.

## Introduction

Can breathwork enhance connection to self, others, and nature, thereby supporting individual, collective, and planetary wellbeing? As past reviews have primarily

**Data availability statement:** No datasets were generated or analysed during the current study. All relevant data from this study will be made available upon study completion.

**Funding:** The author(s) received no specific funding for this work.

**Competing interests:** The authors have declared that no competing interests exist.

focused on the individual, there is a need for a broader synthesis that considers these interconnected dimensions. Given today's global challenges, often described as a 'polycrisis' [1–3], it is essential to explore the potential of breathing interventions to promote wellbeing relating to individuals, the collective, and planet. For instance, as we discuss below, breathing interventions may support the 'inner' development required to facilitate the 'outer' change needed to address the polycrisis and promote systemic change.

Encompassing a range of techniques, breathwork, such as prāṇāyāma (yogic breath control or extension), have been practiced for millennia [4]. Despite historical significance, scientific interest in the potential health and wellbeing benefits is relatively recent [5], with the number of studies referencing 'breathing technique' or 'breathing intervention' in Scopus more than doubling in the last decade (search date: 18th February 2025). So far, research interest has largely focused on individualised outcomes such as affect [6], physiological and psychological stress [7], blood pressure, heart rate [8], and heart rate variability (HRV) [9]. While existing reviews have focused on the effects of breathwork on outcomes aligned to hedonic (feeling good) and eudaimonic (functioning well) wellbeing perspectives [e.g., 10, 11], they typically neglect wider determinants. This individualistic focus reinforces neoliberal perspectives and isolates wellbeing from interconnected social and ecological systems [12], contributing to societal challenges such as loneliness [13] and the climate crisis [14], and undermining wellbeing [15,16]. Addressing these crises requires a shift away from individualistic models toward a more integrated, holistic understanding of wellbeing. This review therefore aims to synthesise existing literature on breathing interventions from a holistic perspective, considering their impact on self, others, and the natural world.

## Breathwork

Breathwork, defined as "exercises, techniques, and therapies that involve manipulating the manner in which one breathes" [17], have deep cultural and spiritual roots often emphasising the interconnectedness of all living beings. Techniques such as qigong (Chinese breath, movement and mind practices to cultivate vital energy) [18], g-tummo (Tibetan practice to control inner energy) [19], prāṇāyāma [20], and shamanic breathing [21] originate from traditions that foster reverence for nature (e.g., Hinduism, Buddhism, Taoism, and Shamanism) [22]. For instance, prāṇāyāma, derived from yoga, conceptualises prāṇā as a fundamental life force, with yāma meaning control or extension [20]. In contemporary practice, breathwork is often integrated with nature-based approaches such as forest bathing (shinrin-yoku) [23] and cold-water exposure (Wim Hof Method; WHM) [24,25], further highlighting its potential role in promoting planetary wellbeing, a concept described later.

While breathwork is well-investigated for stress reduction [e.g., 7, 26], HRV enhancement [e.g., 9, 27], and increasingly hedonic and eudaimonic indicators of wellbeing [e.g., 10, 28], its effects may extend beyond individual benefits. For example, evidence suggests that prāṇāyāma strengthens social connectedness [29], and fosters a deeper relationship with the body through improved interoception and

emotional regulation [28]. Body awareness, in turn, is a key predictor of nature connection [30]. Sudarshan Kriya Yoga (SKY), incorporating prāṇāyāma, has been associated with nature connection and climate change awareness [31], while WHM may promote holistic wellbeing through enhanced connection to self, others, and nature [25]. Similarly, holotropic breathwork has been linked to increased capacity to connect with others, sensitivity to one's own needs, and greater awareness of one's environment [32]. Despite these indications, no comprehensive review has considered a broader, holistic perspective of the wellbeing-related impacts of breathing interventions.

## Defining wellbeing

Wellbeing is a contested concept, with definitions that are often contradictory, sometimes overlapping, and overly focused on the individual [33–36]. Hedonic models emphasise subjective experiences of positive emotions and life satisfaction [37], while eudaimonic perspectives highlight meaning [38,39], purpose [40], autonomy [40,41], and social relationships [40,42] (Table 1). Models of flourishing have sought to integrate these perspectives [e.g., 38, 39, 45], although they sometimes contradict each other; for instance, the pursuit of meaning can come at the expense of positive emotion [66], highlighting the complexity of the construct of wellbeing [67]. Historically, wellbeing models have centred on individual experiences while neglecting broader social and environmental determinants [35,63]. However, newer conceptualisations acknowledge that individuals do not 'feel good' or 'function well' in isolation [34,36,56,64,68].

A systems perspective of wellbeing emphasises interconnectedness and interdependence [36,56,64,68,69] recognising that human wellbeing is inseparable from that of the planet [56,69,70]. Holistic wellbeing frameworks extend beyond the individual to incorporate social and environmental dimensions [e.g., 67, 56, 64, 65, 71]. Proponents of this approach call for more inclusive measures [e.g., 72], emphasising factors such as connection, vagal health [67,64,65], nature (e.g., biophilia), social trust and happiness [72], and societal harmony [56]. Emerging models such as the GENIAL framework [34,67,64,65] and the model of 'sustainable wellbeing' [56], span multiple domains (systems) within which wellbeing can arise (e.g., self, others, and nature), reinforcing the interconnectedness of individual, collective, and planetary domains of wellbeing.

Through this lens, individual wellbeing is defined as feeling good and functioning well [73], while collective wellbeing involves interdependently feeling good and functioning well [74]. Planetary wellbeing (also referred to as ecological wellbeing) highlights the reciprocal relationship between humans and the environment [51,70], and has been described as "the highest attainable standard of wellbeing for human and non-human beings and their social and natural systems" [75 p.4]. Measures of planetary wellbeing include pro-environmental behaviour [e.g., 76], nature connection [e.g., 77], and indices of 'environmentally sustainable' wellbeing [e.g., 57–62, 78]. Importantly, interventions that enhance self-awareness [e.g., 79, 80], foster belonging [e.g., 81], and deepen nature connection [e.g., 82] may promote wellbeing across all three domains.

## An urgent need for a more holistic perspective

The escalating global crises (e.g., climate change, war, political instability, and mental health challenges) [1–3,14,83,84] underscore the urgency of a more holistic approach to wellbeing. Despite policy efforts such as the United Nations Sustainable Development Goals, progress remains inadequate [85]. Dominant neoliberal ideologies, which emphasise individualism and economic growth, contribute to societal fragmentation and ecological degradation [86–89]. Addressing these crises requires reflection on collective and planetary perspectives, with increasing recognition of the need to integrate 'inner' development with external solutions [89,90].

Current reviews of breathing interventions reinforce an individualistic outlook, overlooking their potential to foster connection across multiple domains. In this regard, breathwork may serve as a means of inner transformation, for example through strengthening mind-body awareness and re-establishing relationships with others and the environment [67,87,89]. A scoping review is a necessary next step to explore these possibilities, allowing for a more inclusive conceptualisation of wellbeing that accounts for individual, collective, and planetary dimensions. Furthermore, preliminary literature searches

**Table 1. Summary of theories, models and measures contributing to a holistic wellbeing definition.**

| Theory, model or measure | Included components | Domain |
|---|---|---|
| Subjective Wellbeing [37] | Subjective assessment of positive affect, negative affect, life satisfaction. | Individual |
| Psychological Wellbeing Theory [40,43] | Autonomy, environmental mastery, personal growth, positive relations with others, purpose in life, self-acceptance. | Individual, Collective |
| Flourishing [38,39] | Positive emotions, engagement, relationships, meaning/mattering, accomplishment. | Individual, Collective |
| Self-determination theory [44] | Basic psychological need for autonomy, relatedness and competence. | Individual, Collective |
| Social Wellbeing [42] | Social coherence, social actualisation, social integration, social contribution, social acceptance. | Individual, Collective |
| Flourishing [45] | Emotional wellbeing, psychological wellbeing, social wellbeing. | Individual, Collective |
| Flourishing [46] | Competence, emotional stability, engagement, meaning, optimism, positive emotions, positive relationships, resilience, self-esteem, vitality. | Individual, Collective |
| Eudaimonic Wellbeing [47] | Self-discovery, perceived development of one's best potentials, sense of purpose and meaning in life, intense involvement in activities, investment of significant effort, enjoyment of activities as personally expressive. | Individual |
| Biophilia Hypothesis [48] | An innate affinity to seek connection with nature and other forms of life. | Individual, Planetary |
| Psycho-evolutionary theory [49] | Exposure to nature can generate automatic positive affect. | Individual, Planetary |
| Spiritual wellbeing [50] | Personal wellbeing (e.g., self-awareness, joy), communal wellbeing (e.g., trust, respect others), transcendental wellbeing (e.g., relation with divine), environmental wellbeing (e.g., one with nature, value in nature). | Individual, Collective, Planetary |
| Ecological wellbeing [51] | Wellness of the ecological system, and more generally planet earth, encompassing the relationship between humans and their environment. | Individual, Planetary |
| Sustainable wellbeing [52] | Integrating wellbeing of individual, other members of society and natural environment. | Individual, Collective, Planetary |
| Sustainable happiness and wellbeing [53–55] | Happiness that contributes to individual, community, and/or global wellbeing without exploiting other people, the environment, or future generations. | Individual, Collective, Planetary |
| Sustainable wellbeing [56] | All systems and wellbeing domains are in balance and harmony including the individual (physical, mental, social, spiritual dimensions), self-and-other, people-and-environment and over time. | Individual, Collective, Planetary |
| Happy Planet Index [57–62] | Global index of sustainable wellbeing. | Individual, Planetary |
| GENIAL framework [63,64,65] | Genomics, Environment, vagus Nerve, social Interaction, Allostatic regulation, Longevity. | Individual, Collective, Planetary |

suggests an emerging research focus on the social and environmental impacts of breathwork [e.g., 25, 29, 31], highlighting the need for synthesis and identifying gaps in the existing knowledge base.

## Methods

The scoping review will be conducted in accordance with the Joanna Briggs Institute (JBI) manual for scoping reviews [91] and will conform to the reporting standards published in the Preferred Reporting Items for Systematic reviews and Meta-Analyses extension for Scoping Reviews (PRISMA-ScR) [92]. The current protocol has been registered on the Open Science Framework (osf.io/j7phg), and its development was guided by the Preferred Reporting Items for Systematic Review and Meta-Analysis Protocols (PRISMA-P; S4 Table) [93]. The methodology for a scoping review was also guided

by a six-stage framework described by Arksey and O'Malley [94] including: (1) the identification of a research question, (2) identifying relevant studies, (3) study selection, (4) charting the data, (5) collating, summarising, and reporting the results, and (6) consultation with breathwork facilitators. The optional sixth stage is conducted in parallel with earlier stages, and follows Buus et al. [95] recommendations for consultation exercises. The scoping review process began in April 2025 and is anticipated to take approximately twelve months to complete.

## Stage 1: Identifying the research question

Research questions were developed in line with the JBI [91] recommended framework, including a focus on population, concept and context (Table 2). The primary research question is:

How do breathing interventions promote holistic wellbeing?

Additional questions that will be answered in the scoping review will include:

1. What types of breathing interventions have been used?

2. How have they been delivered?

3. What is the duration (number of minutes per session) and frequency (sessions per week) of practice?

4. What are the reported outcomes relevant to holistic wellbeing (S3 Table)?

5. What populations have been studied?

6. What are the mediating and moderating effects of breathing interventions on holistic wellbeing?

Answering these questions will help inform further research, including the design and delivery of novel breathing interventions, which consider broader socio-ecological contexts and support wellbeing relating to self, others, and planet.

## Stage 2: Identifying relevant studies

**Search strategy.** The search strategy attended to established guidelines for conducting scoping reviews [91,94], and focused on identifying relevant quantitative and qualitative research in the published literature. The grey literature will also be reviewed for relevant studies. To ensure "in-depth and broad" retrieval of relevant literature [94, p.22], an extensive list of search terms was collated through several consultation exercises. Firstly, drawing on our knowledge of breathwork and wellbeing science and in collaboration with a skilled librarian from Universidade Católica Portuguesa, an initial search of PubMed and APAPsycArticles was conducted to identify articles related to the key themes of breathing interventions and wellbeing. Search terms relating to various breathing techniques and concepts relating to holistic wellbeing were selected (S3 Table). Relevant terms from publications' titles, abstracts, and bibliographies were added to the list. Search terms were expanded following consultation of the APAPsycArticles thesaurus. Google Search and Google Scholar were then explored to capture the variety of references to breathing interventions, further developing the search term list. Terms which consistently returned studies outside of scope (e.g., 'breathing' returned multiple pulmonary-related studies) were removed or edited to be more specific. The full list of search strategy terms can be viewed in S3 Table.

**Table 2. The population, concept and context relating to the primary research question.**

| Population | Concept | Context |
|---|---|---|
| Individuals or groups of the general population exposed to a breathing intervention. | Individual- or group-based breathing interventions intending to promote indicator(s) of wellbeing, broadly defined, spanning individual, collective and/or planetary domains. | No restrictions will be placed on the context. |

The librarian was then consulted to help determine appropriate databases to search. The following databases were identified: PubMed, Scopus, Web of Science, APAPsycArticles, Psychology and Behavioural Sciences Collection and Cochrane. Grey literature sources will include PsyArXiv, ProQuest and Google Scholar. Reference lists will also be screened for additional studies relevant to breathing interventions and indicators of holistic wellbeing. The search strategy (S1 Appendix), including all identified keywords and index terms, will be adapted accordingly for each database and information source utilised (see S2 Table for examples of database-specific search strategies). No limits will be placed on language or date range. For identified non-English language evidence, translation services such as DeepL translator tool [96] will be used to determine whether inclusion criteria are met. Translations will be reviewed by individuals fluent in the relevant language. Any evidence that cannot be translated will be classified as "Studies awaiting classification" and reflected in the PRISMA flow diagram [97] in the full review [98]. Following the search, all identified citations will be collated and uploaded into Rayyan [99].

### Eligibility criteria

**Population.** The review will include studies of the general population which focus on the promotion of wellbeing, rather than reduction of ill-being. Therefore, specific sub-groups for which breathing interventions are delivered to manage symptoms as part of a healthcare treatment plan (e.g., for asthma or chronic obstructive pulmonary disease) will be excluded.

**Concept.** Within the current scoping review, there are two main concepts of interest. The first is breathing interventions, or breathwork, as defined above, which must include interventions that deliberately manipulate the breath of the target population. This will include isolated breathing interventions (e.g., prāṇāyāma), and will not include studies that also test other complementary components (e.g., yoga asana), unless the breathing component is tested individually, or its effects can be isolated. This is because these additional components may influence the outcome measures. See S5 Table for a non-exhaustive selection of prominent breathing interventions, chosen based on the first author's experience and as potential examples that may be identified in the search. The second concept is holistic wellbeing, informed by socio-ecological models of wellbeing such as the GENIAL framework [64,65]. Therefore, studies must also include outcome measures related to at least one of the domains within a broader conceptualisation of wellbeing spanning multiple domains (S3 Table). This conceptualisation spans the individual (e.g., positive affect, self-connection), community (e.g., perception of connectedness to others, social cohesion) and broader ecosystems (e.g., nature connection, pro-environmental attitudes).

**Context.** We will not limit the context for the current scoping review. All studies (published or in the grey literature) will be included regardless of geographic location, specific settings, or time of publication.

**Types of sources.** This review will consider all quantitative and qualitative, published, and unpublished primary research studies and evidence syntheses (e.g., systematic reviews, meta-analyses, other scoping reviews). Evidence syntheses will be included alongside original research in order to facilitate the identification of as many primary studies as possible. There will be no restrictions placed on study design or methodology. Where there are duplications due to inclusion of a synthesis and a primary source cited in that synthesis, the primary source will be excluded unless that data is missing [91]. The full text must be accessible by the authors. If full access is unavailable institutionally, the corresponding author will be contacted via email (from an institutional email) up to a maximum of three attempts. One month will be provided for authors to respond, with this date included in all email contacts. Two weeks after the initial attempt, a follow up email will be sent and subsequently after another week. Criteria for inclusion and exclusion are presented in Table 3.

### Stage 3: Study selection

After duplicate citations have been removed, the remaining titles and abstracts will subsequently be screened in a pilot exercise, as recommended by JBI guidelines [91,100]. This will allow for refinement within the review team and consist of

**Table 3. Types of sources eligibility criteria.**

| Criterion | Inclusion | Exclusion |
|---|---|---|
| Type of study | Published and unpublished primary research studies and evidence syntheses reporting on wellbeing-based outcomes targeted by breathing interventions. This includes quasi-experimental designs, randomised controlled trials, interviews, focus groups and observations. | Non-intervention based research. |
| Source | Studies located in a peer-reviewed journal, available as an e-thesis or on a grey literature website. | Studies not located in a peer-review journal, available as an e-thesis or on a grey literature website. |
| Availability | Full text available. | No full text available. |
| Study focus | Use of at least one breathing intervention. | Studies do not include at least one breathing intervention or are also testing other complementary components, unless the breathing component is tested individually, or its effects are isolated. |

reviewing a subset of potentially relevant titles and abstracts. A sample of 25 titles/abstracts will be selected randomly and reviewed by FEB and EB. These will be screened using the eligibility criteria and the reviewers will discuss the chosen papers and any required amendments to the criteria. Any revisions of the eligibility criteria, search terms or methodology will be reported in the final review. When 75%+agreement is achieved, full screening will commence. Full screening will consist of two reviewers, FEB and EB, independently assessing all titles and abstracts against the final eligibility criteria. Utilising the 'blinding' functionality in Rayyan, reviewers' decisions will not be visible, and conflicts will be identified with opportunities to discuss decisions in real time. Any uncertainties or discrepancies will be discussed between the two reviewers, and a third reviewer called upon should they require assistance breaking a tie. Subsequently, following a similar screening process, the two reviewers will independently assess the full articles against eligibility criteria. The search may be altered and expanded during this process to account for potential new discoveries [101] due to the broad nature of scoping reviews, and focus on the two broad concepts. As recommended by Tricco et al. [92], results of the process and rationale for excluding studies will be reported using a Rayyan-generated PRISMA flow diagram [97] in the full review. Excluded full-text sources, along with reasons for exclusion, will be included in an appendix.

## Stage 4: Charting the data

The data extraction form (Table 4) will be pilot tested by FEB and EB on a random selection of 10 papers [102]. This process will assess consistency and reliability in extraction as well as form functionality [103]. Subsequently, both reviewers will convene to discuss any refinement required of the form. The full data extraction process will involve regular discussion between the two independent reviewers and utilise the finalised form. Extracted data will include general study details as well as population, concept, and context information relevant to the current research question. Breathing interventions will be categorised into slow- and fast-paced techniques (S5 Table). In the eventuality that any information is missing or unclear, as with the full text sourcing process, the corresponding author will be contacted via email a maximum of three times requesting clarity or the absent information. If information is not recovered, it will be coded as "Not reported" or if it remains unclear it will be coded as "Unclear" and reported in the full review. In cases of multiple reports of the same study or 'friend studies', these will be treated on a case-by-case basis and the most appropriate strategy employed accordingly. All strategies employed will be detailed when reporting the full review. Data extraction is expected

**Table 4. Data extraction form.**

| Study Details | Population | Breathing Intervention Description | Wellbeing Outcomes | Results | Notes |
|---|---|---|---|---|---|
| Author(s)<br>Title<br>Publication year<br>Aim(s)<br>Study design<br>Recruitment method<br>Field of study, first author<br>Country | Target population<br>Participant characteristics:<br>• Sample size<br>• Age range<br>• Mean age<br>• Sex<br>• Gender identity<br>• Ethnicity<br>• Race<br>Control<br>Other population notes | *For each intervention:* Breathing technique name<br>Technique description/components<br>Categorisation of intervention (fast/slow)<br>Method of delivery<br>Provider of delivery<br>Frequency<br>Duration<br>Location of delivery<br>Total number of intervention groups<br>Contraindication(s) (where applicable)<br>Comparison group(s) (presence/absence and type) | *For each wellbeing outcome of interest:*<br>Wellbeing-related outcome<br>Measures used<br>Timeline of measures<br>Categorisation of outcomes by wellbeing domain | Description of key findings | Limitations of study as identified by authors<br>Relevant quotes (qualitative) |

to increase in speed as familiarity in the process grows. The two reviewers will engage in regular discussion throughout the process to highlight any issues. A third reviewer will be on hand to resolve disagreement. Any alterations made during the process will be disclosed in the final review.

## Stage 5: Collating, summarising and reporting the results

Once data has been extracted, all authors will convene to discuss the data, plan analysis and reporting. We intend to present a quantitative analysis of the extent, nature, and distribution of the studies included in the review using tables and charts, as well as a narrative synthesis. Using the multiple wellbeing domains presented in the GENIAL framework [64,65] as a guide, from the charted data (Table 4) we will produce tables, charts, and visualisations presenting the distribution of evidence by grouping outcomes relating to these domains. Primary outcomes of interest will be focused on any wellbeing measure that could be defined holistically (e.g., Happy Planet Index [62] or Gallup World Poll [e.g., 72]). Secondary outcomes will be focused on measures that can be categorised within the domains of individual, collective, and planetary wellbeing. Further, due to the volume of techniques and potential homogeneity of outcomes in relation to speed (e.g., slow-paced techniques can produce relaxation-based outcomes [e.g., 11], while fast-paced can energise [e.g., 104]), we will also organise breathing interventions by pace (e.g., slow- and fast-paced) similar to existing literature [e.g., 5, 11]. To answer the research questions, we will summarise (1) the types and characteristics of breathing interventions studied, (2) wellbeing outcomes and their direction (beneficial, adverse, mixed, no difference/unchanged) reported within each domain, (3) wellbeing-related measures used, (4) populations studied, (5) methodological approaches employed, and (6) the key conclusions drawn from evidence syntheses. Quantitative analysis will comprise of producing tables and charts showing the frequencies of studies based on this data (e.g., breathing intervention studies by wellbeing domain, as exemplified in Table 5 and Fig 1), aiming to highlight key areas of interest and illuminate gaps in the literature for future research.

We will also conduct a narrative synthesis to understand the effectiveness of breathing interventions in relation to the presented broader conceptualisation of wellbeing, why and how this might occur, and the degree to which existing breathing intervention research utilises this conceptualisation. Data analysis and presentation details will be reported fully in the final review.

## Stage 6: Consultation

A consultation exercise will be conducted utilising the optional sixth stage [94]. Initial findings from Stage 5 will be shared with breathwork facilitators with two main aims (1) to identify gaps not explored in the literature and (2) seek

**Table 5. Illustrative, hypothetical example of a results table, reporting frequency of breathing intervention studies by pace, wellbeing domain and direction of outcome.**

| | | Wellbeing Domain | | | | | | | | | | | |
|---|---|---|---|---|---|---|---|---|---|---|---|---|---|
| | | Individual | | | | Collective | | | | Planetary | | | |
| | | + | − | +/- | O | + | − | +/- | O | + | − | +/- | O |
| **Breathing technique by pace** | **Slow-paced [Total n studies]** | | | | | | | | | | | | |
| | Breathing intervention 1 (e.g., Nadi Shodhana) | | | | | | | | | | | | |
| | ... | | | | | | | | | | | | |
| | **Fast-paced** | | | | | | | | | | | | |
| | Breathing intervention 5 (e.g., Conscious Connected Breathing) | | | | | | | | | | | | |
| | ... | | | | | | | | | | | | |
| | **Both (slow- and fast-paced)** | | | | | | | | | | | | |
| | Breathing intervention 9 (e.g., g-tummo) | | | | | | | | | | | | |
| | ... | | | | | | | | | | | | |

Key: **+** = beneficial effect, **-** = adverse effect, **+/-** mixed results, **O** = no difference/unchanged.

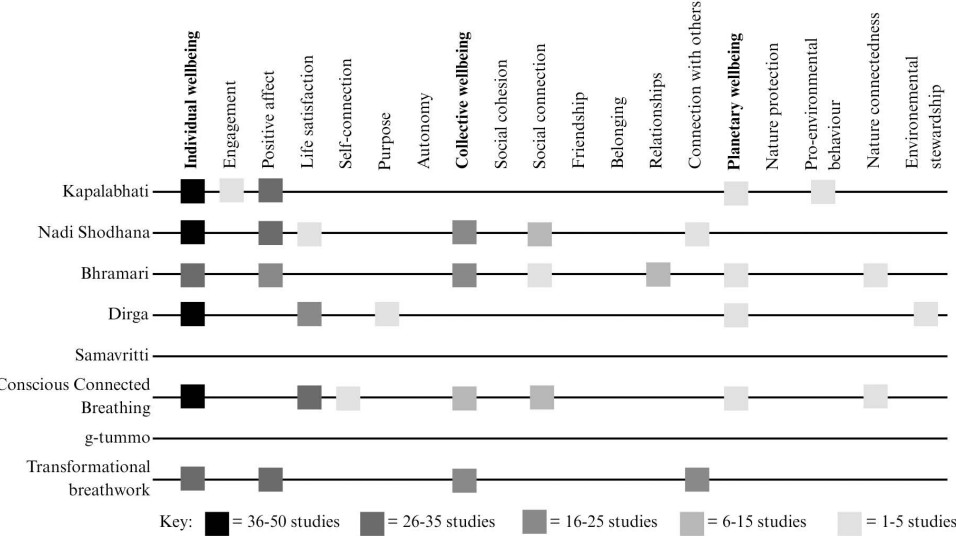

**Fig 1. Illustrative, hypothetical example of a visualisation reporting on frequency of breathing intervention by wellbeing outcome.**

feedback on how to discuss the findings and implications of the review in an applied context. Potential implications will include three elements (a) opportunities to promote holistic wellbeing outcomes in existing practices, (b) considerations required for this, and (c) recommendations for future research. Breathing interventions have a long, diverse, applied, and largely non-scientific history, suggesting there may be knowledge gaps in the existing literature. It is therefore anticipated that consulting breathwork facilitators will enable additional meaningful and applied perspectives, aligning to Buus et al. [95] and Mak and Thomas [105] critique and recommendations for consultation exercises. Facilitators of a variety of breathing intervention doctrines will be contacted via email and asked whether they would like to consult in the research. We will provide detailed information about the research and what to expect should they wish to assist. The consultation process, associated outcomes, and how they opposed (or not) conventional viewpoints will be included in the full review [95].

## Ethics and dissemination

Ethical approval will not be required as this will be a review of previously published data. Additionally, the consultation made with breathing intervention facilitators is intended to only inform research findings and thus the research carried out 'with' them, rather than 'about' them. As their contribution is therefore deemed as public involvement, ethical approval will not be required [106]. The full scoping review will be published in peer-reviewed journals and study findings may be shared at scientific conferences and with those taking part, as defined in Stage 6 of the scoping review.

## Conclusion

Results of this scoping review will comprehensively report on what is known from the existing literature about how breathing interventions might promote holistic wellbeing. A holistic wellbeing framework will be utilised to help understand how such interventions may enhance wellbeing within each of these domains. In doing so, this scoping review aims to provide an urgently needed overview which looks both at and beyond the individual, also considering the impact of these cost-free, accessible, and sustainable interventions on broader social and environmental outcomes. This will enable a better understanding of how such interventions might be leveraged in support of inner development and rapidly evolving global challenges facing humanity.

## Supporting information

**S1 Appendix.  Search strategy.**
(DOCX)

**S2 Table.  Database-specific search strategies.**
(DOCX)

**S3 Table.  Study concepts and associated search terms.**
(DOCX)

**S4 Table.  PRISMA-P 2015 Checklist.**
(DOCX)

**S5 Table.  Non-exhaustive selection of potential breathing interventions.**
(DOCX)

## Acknowledgments

The search strategies and databases selected for the current scoping review were developed with the valuable advice of an experienced librarian from Universidade Católica Portuguesa. Accordingly, the authors would like to extend our appreciation to Maria Perdigão for her comprehensive support and guidance in database-specific search strategies.

## Author contributions

**Conceptualization:** Fern Eleanor Beauchamp, Andrew H Kemp.

**Data curation:** Fern Eleanor Beauchamp.

**Formal analysis:** Fern Eleanor Beauchamp.

**Investigation:** Fern Eleanor Beauchamp.

**Methodology:** Fern Eleanor Beauchamp.

**Project administration:** Fern Eleanor Beauchamp.

**Resources:** Fern Eleanor Beauchamp.

**Supervision:** Iolanda Costa Galinha, Andrew H Kemp.

**Validation:** Fern Eleanor Beauchamp, Iolanda Costa Galinha, Andrew H Kemp.

**Visualization:** Fern Eleanor Beauchamp.

**Writing – original draft:** Fern Eleanor Beauchamp.

**Writing – review & editing:** Fern Eleanor Beauchamp, Emily Bispo, Iolanda Costa Galinha, Andrew H Kemp.

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
