## [Decision Letter · Decision Letter 0]

26 Jun 2025

Dear Dr. Kemp,

Thank you for submitting your manuscript to PLOS ONE. After careful consideration, we feel that it has merit but does not fully meet PLOS ONE’s publication criteria as it currently stands. Therefore, we invite you to submit a revised version of the manuscript that addresses the points raised during the review process.

We look forward to receiving your revised manuscript.

Kind regards,

Hidetaka Hamasaki

Academic Editor

PLOS ONE

Journal Requirements:

Reviewers' comments:

Reviewer's Responses to Questions

**Comments to the Author**

1. Does the manuscript provide a valid rationale for the proposed study, with clearly identified and justified research questions?

Reviewer #1: Yes

Reviewer #2: Yes

2. Is the protocol technically sound and planned in a manner that will lead to a meaningful outcome and allow testing the stated hypotheses?

Reviewer #1: Partly

Reviewer #2: Yes

3. Is the methodology feasible and described in sufficient detail to allow the work to be replicable?

Reviewer #1: Yes

Reviewer #2: Yes

4. Have the authors described where all data underlying the findings will be made available when the study is complete?

Reviewer #1: No

Reviewer #2: Yes

5. Is the manuscript presented in an intelligible fashion and written in standard English?

Reviewer #1: Yes

Reviewer #2: Yes

You may also provide optional suggestions and comments to authors that they might find helpful in planning their study.

Reviewer #1: The protocol is well written but it will benefit from a few amendments

1: The registration number obtained when the protocol was registered in open science framework must be stated in the manuscript

2: I wonder if a meta-analysis will be done after the scoping review, if so then perhaps a subgroup analysis and sensitivity analysis must be done and must be stated in the protocol.

Reviewer #2: I found the topic to be highly relevant and timely, and your protocol is generally well-structured and comprehensive. The detailed approach you propose for your scoping review suggests a thorough and systematic investigation into breathwork and its impact on holistic wellbeing. I appreciate the effort put into developing such a detailed plan.

My comments are provided below, starting with the more overarching and methodological points, followed by more specific suggestions to further enhance the clarity, rigor, and conciseness of your protocol.

Major Comments

1. Introduction Length and Focus: For a protocol, the Introduction section appears quite extensive. While providing context is crucial, some of the more detailed background information and extensive justification for the review's necessity might be more suitable for the full scoping review manuscript itself. I suggest considering a more concise and direct approach for the protocol's introduction, focusing primarily on setting the stage for why this protocol is needed.

2. Search Strategy and Disease-Specific Terms: I noted that your search strategy includes specific disease terms (e.g., "cancer[Title] OR patient*[Title] OR diabetes[Title] OR COPD[Title] OR asthma[Title] OR disease*[Title] OR cystic fibrosis[Title] OR myocardial infarction[Title] OR hypertension[Title] OR treatment[Title]"). Given that the title of your review is "Breathwork and holistic wellbeing," which suggests a broad exploration of the topic, the specific inclusion of these diseases in the search terms requires further clarification and justification within the protocol. It is not clear why these specific conditions have been chosen over others, or if this implies a narrower scope than initially suggested by the title.

Please clarify the rationale: Are these diseases chosen because breathwork interventions are predominantly studied in these populations, or do they represent an intentional limitation of your review's scope?

Implications for Scope: If your intention is to focus solely on these conditions, this should be explicitly stated in the protocol's eligibility criteria, and potentially even reflected in the title to align expectations.

Risk of Omission: Without a clear justification, there's a risk of inadvertently excluding relevant studies on breathwork in other health conditions or general populations, thereby limiting the comprehensiveness typically associated with a scoping review. This point is crucial for ensuring the transparency and rigor of your chosen search methodology.

3. Clarification on "Evidence Syntheses" and Inclusion Criteria:

In lines 269-270, you state that the review "will consider all quantitative and qualitative, published and unpublished primary research studies and evidence syntheses." It would be highly beneficial to explicitly define what you mean by "evidence syntheses" in this context. Is this referring to other types of reviews (e.g., systematic reviews, meta-analyses, or other scoping reviews)?

Furthermore, if "evidence syntheses" refer to other reviews, it's important to clarify your rationale for including them alongside primary studies. While existing reviews can be valuable for identifying primary studies or for comparison, relying on their data synthesis or interpretations might introduce bias if their methodologies or eligibility criteria do not perfectly align with your own. Please clarify if you intend to extract data from these "syntheses" or primarily use them to identify primary studies. A clear justification for this approach, particularly regarding how potential biases from secondary data selection/interpretation will be mitigated, would strengthen the protocol.

4. Explicitly Stating Outcomes of Interest: While you mention following the GENIAL framework (Stage 5), it would significantly enhance the clarity and reduce potential bias during the review process if you could provide more detail regarding the specific types of outcomes you anticipate identifying and extracting from the included articles. Outlining the primary and secondary outcomes of interest within the protocol would provide a clearer roadmap for the subsequent review.

5. Rationale for Fixed Pilot Test Sample Sizes: In line 289, you specify a pilot test of 25 titles, and in line 309, a pilot data extraction from 10 articles. Could you clarify the rationale for these fixed numbers? It might be more robust to define a percentage of the total number of identified titles/abstracts or articles, as the absolute numbers might vary significantly depending on the initial literature search yield.

6. Relevance and Placement of Tables:

Table 2 (Timeline): The inclusion of a detailed timeline for the project (Table 2) does not appear to add significant methodological value to the protocol itself. It might be more appropriate for project management documentation rather than a scientific protocol.

Table 4 & Table 5: Given their size and detailed nature, I suggest considering moving Table 4 and Table 5 to supplementary materials. This would enhance the readability and flow of the main protocol document while still making the detailed information accessible to readers.

Table 6: This table seems largely redundant, as the information presented is already clearly outlined in the text. You could consider removing it entirely or, alternatively, condensing the surrounding text and presenting the information exclusively within the table.

Specific Points and Minor Suggestions

• Introduction References: There are instances in the Introduction (e.g., lines 62-64) where concepts are presented without corresponding references. I recommend a thorough review of the Introduction to ensure all claims and conceptual foundations are adequately supported by citations.

• Sub-title Rephrasing (Line 105): At line 105, the section title is presented as a question; I suggest rewording to be more objective and declarative.

• Use of Dashes: Throughout the manuscript, consider replacing long dashes used to interject phrases with commas. This can often improve textual clarity and objectivity.

• Reference Management: The sequencing of references in lines 164-166 (reference 93 immediately after 91) and then reference Buus et al. also identified as 93, in line 171 suggests that a reference management software (such as Zotero, Mendeley, or EndNote) might not be consistently used. I highly recommend utilizing such a tool to prevent these types of discrepancies and to ensure consistent style and accurate numbering, especially given the anticipated large number of references for both this protocol and the eventual scoping review.

• Table References in Text: When referring to tables in the text, it is not necessary to include the word "see" (e.g., "Table 1" is sufficient instead of "see Table 1").

• Table 5 Details: For Table 5, if it's not exhaustive, it would be helpful to briefly explain the rationale behind selecting these specific examples over others. Additionally, it would be beneficial to include references for the different techniques presented and their descriptions within the table itself.

• Table 7 (Data Extraction Fields):

Population Characteristics: Regarding the "Population" section, please clarify the reason/relevance of distinguishing between "sex" and "gender identity" for data extraction. Also, while "nationality" is listed, extracting the "country/location where the study was conducted" (which you already have) might be more straightforward and consistently available than participants' nationalities. When referring to "ethnicity," please also include "race," as these are distinct concepts.

Control Group Data: It's unclear whether you intend to collect the same level of detailed demographic and intervention-related data for the control groups or simply note their presence/absence. Please clarify this for the "Comparison group(s)" column within the "Breathing Intervention Description" section.

• Clarity on "Homogeneity of Outcomes" (Lines 336-337): In lines 336-337, you state, "Further, due to the volume of techniques and potential homogeneity of outcomes in relation to speed…" Could you please clarify if "homogeneity of outcomes" is an expected characteristic or if this might be a typographical error? If homogeneity is expected, a brief explanation would be useful.

• Figure 1 and Table 8 (Illustrative Formats):

While you aptly present Figure 1 and Table 8 "as exemplified" to illustrate potential reporting formats for your results, please ensure that Figure 1 is clearly cited if it is adapted from an existing source. If it is an original, hypothetical example created for this protocol, it would be beneficial to state this explicitly. For a protocol, whose primary purpose is to outline the methodology, committing to an exact visual format for presenting findings can sometimes be premature. The optimal presentation strategy often evolves during the data synthesis phase of a scoping review, especially given the potential heterogeneity of the included literature. You might consider adding a brief note to reinforce that these are illustrative examples or potential formats, allowing for flexibility in the final representation of results.

I trust these comments will assist you in refining your protocol.

**Do you want your identity to be public for this peer review?** For information about this choice, including consent withdrawal, please see our Privacy Policy

Reviewer #1: **Yes: ** Irene Boateng

Reviewer #2: No

---

## [Author Response · Author response to Decision Letter 1]

11 Aug 2025

Reviewer #1

Opening comment

Feedback: The protocol is well written but it will benefit from a few amendments.

Response: We are grateful to the reviewer for their review of the manuscript and their kind comments.

Point 1

Feedback: The registration number obtained when the protocol was registered in open science framework must be stated in the manuscript.

Response: Thank you for this suggestion, we have added the DOI to the manuscript obtained when the protocol was registered in Open Science Framework (line 164).

Point 2

Feedback: I wonder if a meta-analysis will be done after the scoping review, if so then perhaps a subgroup analysis and sensitivity analysis must be done and must be stated in the protocol.

Response: A meta-analysis is not currently part of the scoping review. However, the scoping review will determine our next steps and a decision will be made at a later date, on completion of the scoping review.

Reviewer #2

Opening comment

Feedback: I found the topic to be highly relevant and timely, and your protocol is generally well-structured and comprehensive. The detailed approach you propose for your scoping review suggests a thorough and systematic investigation into breathwork and its impact on holistic wellbeing. I appreciate the effort put into developing such a detailed plan.

My comments are provided below, starting with the more overarching and methodological points, followed by more specific suggestions to further enhance the clarity, rigor, and conciseness of your protocol.

Response: We would like to thank the reviewer for their thoughtful and encouraging feedback and appreciate the recognition of our work’s relevance and comprehension. We believe their valuable suggestions have helped improve the manuscript’s clarity and conciseness and further enhanced the quality and rigour of our protocol.

Point 1. Introduction Length and Focus

Feedback: For a protocol, the Introduction section appears quite extensive. While providing context is crucial, some of the more detailed background information and extensive justification for the review's necessity might be more suitable for the full scoping review manuscript itself. I suggest considering a more concise and direct approach for the protocol's introduction, focusing primarily on setting the stage for why this protocol is needed.

Response: We appreciate this helpful suggestion. After careful consideration of the reviewers’ feedback, we have revised the introduction to improve its clarity, focus and length. However, given the emerging nature of the wellbeing perspective underpinning this review, we have taken a measured approach to reducing its length to ensure that sufficient background remains to justify the rationale and significance of our approach. We believe that the revised version strikes a clearer balance between brevity and the necessary context for these novel concepts.

Point 2. Search Strategy and Disease-Specific Terms

Feedback: I noted that your search strategy includes specific disease terms (e.g., "cancer[Title] OR patient*[Title] OR diabetes[Title] OR COPD[Title] OR asthma[Title] OR disease*[Title] OR cystic fibrosis[Title] OR myocardial infarction[Title] OR hypertension[Title] OR treatment[Title]"). Given that the title of your review is "Breathwork and holistic wellbeing," which suggests a broad exploration of the topic, the specific inclusion of these diseases in the search terms requires further clarification and justification within the protocol. It is not clear why these specific conditions have been chosen over others, or if this implies a narrower scope than initially suggested by the title.

Please clarify the rationale: Are these diseases chosen because breathwork interventions are predominantly studied in these populations, or do they represent an intentional limitation of your review's scope?

Implications for Scope: If your intention is to focus solely on these conditions, this should be explicitly stated in the protocol's eligibility criteria, and potentially even reflected in the title to align expectations.

Risk of Omission: Without a clear justification, there's a risk of inadvertently excluding relevant studies on breathwork in other health conditions or general populations, thereby limiting the comprehensiveness typically associated with a scoping review. This point is crucial for ensuring the transparency and rigor of your chosen search methodology.

Response: We thank the reviewer for their comments on this point. For clarification, the disease-specific terms used in the search strategy are for exclusion purposes only. We are using these terms as an automated means to filter out and reduce the number of hits. We added these terms as exclusions because our initial literature searches were returning many studies which included participants with these specific diseases, which were irrelevant to the research question.

Point 3. Clarification on "Evidence Syntheses" and Inclusion Criteria

Feedback: In lines 269-270, you state that the review "will consider all quantitative and qualitative, published and unpublished primary research studies and evidence syntheses." It would be highly beneficial to explicitly define what you mean by "evidence syntheses" in this context. Is this referring to other types of reviews (e.g., systematic reviews, meta-analyses, or other scoping reviews)? Furthermore, if "evidence syntheses" refer to other reviews, it's important to clarify your rationale for including them alongside primary studies. While existing reviews can be valuable for identifying primary studies or for comparison, relying on their data synthesis or interpretations might introduce bias if their methodologies or eligibility criteria do not perfectly align with your own. Please clarify if you intend to extract data from these "syntheses" or primarily use them to identify primary studies. A clear justification for this approach, particularly regarding how potential biases from secondary data selection/interpretation will be mitigated, would strengthen the protocol.

Response: We appreciate the reviewer’s comments and have responded by providing examples of what we mean by “evidence syntheses” (systematic reviews, meta-analyses, other scoping reviews) to lines 263-264. Our rationale for including evidence syntheses alongside primary studies is to help us to locate as many relevant primary studies as possible. We will not be focused on their interpretation or data synthesis and have amended the manuscript (lines 264-265) to include this rationale. In addition, we will include a descriptive summary of the key conclusions drawn from the evidence syntheses we locate, focused on answering our research questions. We have amended the manuscript at lines 336-340 to reflect these changes. We thank the reviewer for their feedback and stimulating further reflection, we believe this has helped strengthen the protocol and the comprehension of the review.

Point 4. Explicitly Stating Outcomes of Interest

Feedback: While you mention following the GENIAL framework (Stage 5), it would significantly enhance the clarity and reduce potential bias during the review process if you could provide more detail regarding the specific types of outcomes you anticipate identifying and extracting from the included articles. Outlining the primary and secondary outcomes of interest within the protocol would provide a clearer roadmap for the subsequent review.

Response: We appreciate the reviewer’s suggestion and have elaborated on the specific types of outcomes that we anticipate identifying and extracting from the included articles. Our primary outcomes of interest will be focused on any wellbeing measure that could be defined more holistically, such as Happy Planet Index [1] or Gallup World Poll [2]). Secondary outcomes will be focused on measures that can be categorised within the domains of individual, collective and planetary wellbeing. We have amended the manuscript accordingly in lines 329-332.

1. Wellbeing Economy Alliance. Happy Planet Index [Internet]. Happy Planet Index. 2021 [cited 2025 Apr 26]. Available from: https://happyplanetindex.org/

2. Lambert L, Lomas T, Weijer MP van de, Passmore HA, Joshanloo M, Harter J, et al. Towards a greater global understanding of wellbeing: A proposal for a more inclusive measure. International Journal of Wellbeing. 2020 May 31;10(2).

Point 5. Rationale for Fixed Pilot Test Sample Sizes

Feedback: In line 289, you specify a pilot test of 25 titles, and in line 309, a pilot data extraction from 10 articles. Could you clarify the rationale for these fixed numbers? It might be more robust to define a percentage of the total number of identified titles/abstracts or articles, as the absolute numbers might vary significantly depending on the initial literature search yield.

Response: We acknowledge the reviewer’s suggestion of defining a percentage of the total number of identified titles/abstracts or articles for piloting. However, our rationale for specifying a pilot test of 25 titles (now line 282) was guided by JBI methodology [3]. Regarding our rationale for specifying a pilot data extraction from 10 articles (now line 301-302), JBI methodology recommends trialling on two to three sources [3,4]. However, we decided to take a more cautious approach to ensure the accuracy of data extraction, as per other institutional recommendations [5].

3. Aromataris E, Lockwood C, Porritt K, Pilla B, Jordan Z, editors. JBI Manual for Evidence Synthesis [Internet]. JBI; 2024 [cited 2025 Jul 31]. Available from: https://synthesismanual.jbi.global/

4. Peters MDJ, Marnie C, Tricco AC, Pollock D, Munn Z, Alexander L, et al. Updated methodological guidance for the conduct of scoping reviews. JBI Evid Synth. 2020 Oct;18(10):2119–26.

5. Western Libraries. Research Guides: Knowledge Synthesis: Systematic & Scoping Reviews: 6. Data Extraction [Internet]. Western University. 2025 [cited 2025 Jul 31]. Available from: https://guides.lib.uwo.ca/knowledgesynthesis/dataextraction

Point 6. Relevance and Placement of Tables

Feedback: Table 2 (Timeline): The inclusion of a detailed timeline for the project (Table 2) does not appear to add significant methodological value to the protocol itself. It might be more appropriate for project management documentation rather than a scientific protocol.

Response: We acknowledge and appreciate the reviewer’s comments of the inclusion of our proposed timeline (Table 2). Accordingly, we have removed this from the manuscript.

Feedback: Table 4 & Table 5: Given their size and detailed nature, I suggest considering moving Table 4 and Table 5 to supplementary materials. This would enhance the readability and flow of the main protocol document while still making the detailed information accessible to readers.

Response: We acknowledge that the manuscript’s readability and flow would be enhanced by moving Table 4 and Table 5 to the supplementary materials (S3 Table and S5 Table, respectively). We have updated the manuscript to reflect these amendments.

Feedback: Table 6: This table seems largely redundant, as the information presented is already clearly outlined in the text. You could consider removing it entirely or, alternatively, condensing the surrounding text and presenting the information exclusively within the table.

Response: We believe structuring the types of sources in tabular form allows for easy reference of the inclusion/exclusion criteria. However, we acknowledge that some elements of the table are redundant as the information is already presented in the text. We have therefore condensed the surrounding text and opted to keep the table (now Table 3) in the main body of the manuscript.

Specific Points and Minor Suggestions

Introduction References

Feedback: There are instances in the Introduction (e.g., lines 62-64) where concepts are presented without corresponding references. I recommend a thorough review of the Introduction to ensure all claims and conceptual foundations are adequately supported by citations.

Response: We have thoroughly reviewed the Introduction to ensure all claims and conceptual foundations are adequately supported by citations. As a result, citations have been added to lines 61, 62 and 156.

Feedback: Sub-title Rephrasing (Line 105): At line 105, the section title is presented as a question; I suggest rewording to be more objective and declarative.

Response: We thank the reviewer for their suggestion and have reworded the section title accordingly.

Feedback: Use of Dashes: Throughout the manuscript, consider replacing long dashes used to interject phrases with commas. This can often improve textual clarity and objectivity.

Response: We thank the reviewer for their suggestion and have replaced long dashes with commas to improve textual clarity and objectivity.

Reference Management

Feedback: The sequencing of references in lines 164-166 (reference 93 immediately after 91) and then reference Buus et al. also identified as 93, in line 171 suggests that a reference management software (such as Zotero, Mendeley, or EndNote) might not be consistently used. I highly recommend utilizing such a tool to prevent these types of discrepancies and to ensure consistent style and accurate numbering, especially given the anticipated large number of references for both this protocol and the eventual scoping review.

Response: We appreciate the diligence in spotting these referencing errors in lines 164-166 and 171. We consistently use Zotero and have ensured that the referencing process has been refreshed and updated.

Table References in Text

Feedback: When referring to tables in the text, it is not necessary to include the word "see" (e.g., "Table 1" is sufficient instead of "see Table 1").

Response: We have amended the manuscript to remove all instances of “see” alongside table references in brackets.

Table 5 Details

Feedback: For Table 5, if it's not exhaustive, it would be helpful to briefly explain the rationale behind selecting these specific examples over others. Additionally, it would be beneficial to include references for the different techniques presented and their descriptions within the table itself.

Response: We acknowledge that a brief explanation of the rationale for selecting the specific breathing interventions and inclusion of references would be helpful. Accordingly, we have included the rationale in the manuscript (lines 246-248) and amended the table (now S5 Table).

Table 7 (Data Extraction Fields)

Feedback: Population Characteristics: Regarding the "Population" section, please clarify the reason/relevance of distinguishing between "sex" and "gender identity" for data extraction.

Response: Our intention to distinguish between "sex" and "gender identity" during data extraction is important because these constructs represent different aspects of identity and may yield different findings. We believe structuring data collection in this way also ensures greater inclusivity and accuracy in analysis.

Feedback: Also, while "nationality" is listed, extracting the "country/location where the study was conducted" (which you already have) might be more straightforward and consistently available than participants' nationalities.

Response: We acknowledge that extracting the “country/location” where the study was conducted might be more straightforward and consistently available, therefore have removed the field “nationality”.

Feedback: When referring to "ethnicity," please also include "race," as these are distinct concepts.

Response: We appreciate the reviewer’s suggestion and have revised the manuscript to include “race” in addition to “ethnicity”, ensuring they are distinct constructs, consistent with APA guidelines. We have amended the data extraction table accordingly (now Table 4).

Control Group Data

Feedback: It's unclear whether you intend to collect the same level of detailed demographic and intervention-related data for the control groups or simply note their presence/absence. Please clarify this for the "Comparison group(s)" column within the "Breathing Intervention Description" section.

Response: We thank the reviewer for their suggestion regarding control group

---

## [Decision Letter · Decision Letter 1]

14 Sep 2025

Breathwork and holistic wellbeing: A protocol for a scoping review.

PONE-D-25-29701R1

Dear Dr. Kemp,

We’re pleased to inform you that your manuscript has been judged scientifically suitable for publication and will be formally accepted for publication once it meets all outstanding technical requirements.

Kind regards,

Hidetaka Hamasaki

Academic Editor

PLOS ONE

Additional Editor Comments (optional):

Reviewer #2:

Reviewers' comments:

Reviewer's Responses to Questions

**Comments to the Author**

1. Does the manuscript provide a valid rationale for the proposed study, with clearly identified and justified research questions?

Reviewer #2: Yes

2. Is the protocol technically sound and planned in a manner that will lead to a meaningful outcome and allow testing the stated hypotheses?

Reviewer #2: Yes

3. Is the methodology feasible and described in sufficient detail to allow the work to be replicable?

Reviewer #2: Yes

4. Have the authors described where all data underlying the findings will be made available when the study is complete?

Reviewer #2: Yes

5. Is the manuscript presented in an intelligible fashion and written in standard English?

Reviewer #2: Yes

You may also provide optional suggestions and comments to authors that they might find helpful in planning their study.

Reviewer #2: Thank you for submitting the revised version of your manuscript PONE-D-25-29701; Breathwork and holistic wellbeing: A protocol for a scoping review.

I would like to acknowledge and commend your diligent work in responding to the suggestions and comments raised during the previous review round. Your thoughtful revisions have significantly enhanced the clarity, rigor, and overall quality of the article, making it a much stronger contribution.

I appreciate your efforts in addressing all points so thoroughly.

Best regards,

**Do you want your identity to be public for this peer review?** For information about this choice, including consent withdrawal, please see our Privacy Policy

Reviewer #2: No

---

## [Editor Report · Acceptance letter]

PONE-D-25-29701R1

PLOS ONE

Dear Dr. Kemp,

I'm pleased to inform you that your manuscript has been deemed suitable for publication in PLOS ONE. Congratulations! Your manuscript is now being handed over to our production team.

Kind regards,

on behalf of

Dr. Hidetaka Hamasaki

Academic Editor

PLOS ONE